# Efficacy, Safety and Immunogenicity of Anti-SARS-CoV-2 Vaccines in Patients with Cirrhosis: A Narrative Review

**DOI:** 10.3390/vaccines11020452

**Published:** 2023-02-16

**Authors:** Konstantina Toutoudaki, Melitini Dimakakou, Theodoros Androutsakos

**Affiliations:** 1First Department of Pediatrics, National and Kapodistrian University of Athens, 11527 Athens, Greece; 2Department of Gynecology and Obstetrics, General Hospital of Chalkida, 34100 Chalkida, Greece; 3Pathophysiology Department, National and Kapodistrian University of Athens, 11527 Athens, Greece

**Keywords:** SARS-CoV-2, COVID-19, vaccines, cirrhosis, neutralizing antibodies, mRNA vaccines

## Abstract

Severe acute respiratory syndrome coronavirus 2 (SARS-CoV-2) causing coronavirus disease 2019 (COVID-19), has led to a pandemic with more than 6.5 million deaths worldwide. Patients with liver cirrhosis (PWLC) are regarded as prone to severe COVID-19. Vaccination against SARS-CoV-2 has been proven to be the most effective measure against COVID-19 and a variety of different vaccines have been approved for use; namely mRNA and vector-based, inactivated, whole virion, and protein subunit vaccines. Unfortunately, only a small number of PWLC were included in phase I–III vaccine trials, raising concerns regarding their efficacy and safety in this population. The authors, in this review, present available data regarding safety and efficacy of anti-SARS-CoV-2 vaccination in PWLC and discuss post-vaccination antibody responses. Overall, all vaccines seem to be extremely safe, with only a few and insignificant adverse events, and efficient, leading to lower rates of hospitalization and COVID-19-related mortality. T- and B-cell responses, on the other hand, remain an enigma, especially in patients with decompensated disease, since these patients show lower titers of anti-SARS-CoV-2 antibodies in some studies, with a more rapid waning. However, this finding is not consistent, and its clinical impact is still undetermined.

## 1. Introduction

Severe acute respiratory syndrome coronavirus 2 (SARS-CoV-2) causing coronavirus disease 2019 (COVID-19) was first described in December 2019 in Wuhan, China [1]. SARS-CoV-2 has, since then, been responsible for almost 650 million cases and more than 6.5 million deaths worldwide [2]. The severity of COVID-19 depends on both the variant of the virus and the host’s comorbidities [3]. Patients with advanced age, diabetes mellitus, hypertension, cardiovascular disease, endocrine, and respiratory diseases are considered to be at high risk for severe COVID-19 [3]. The liver is commonly affected in patients with COVID-19, with severe transaminasemia being present in a substantial number of patients, reaching up to 50% of all SARS-CoV-2 infected individuals. High serum transaminases levels seem to be predictive of poor outcome, however acute liver failure is uncommon in patients with no prior liver disease [4,5]. Patients with chronic liver diseases (CLD) have been recognized as a high-risk group for severe COVID-19 since the beginning of the pandemic [6,7,8,9]. Among them, patients with liver cirrhosis (PWLC), irrespective of the cause, seem to carry the heaviest burden, as shown by an increased morbidity and mortality ratio [6,7,8,9].

Vaccination is the most effective tool against severe COVID-19 and, given the high morbidity and mortality of the pandemic, many different anti-SARS-CoV-2 vaccines, broadly divided into mRNA, inactivated or weakened virus, viral vector, and protein subunit ones, have been licensed with fast-track authorization [10,11,12]. During the months following the initiation of global vaccination, these vaccines proved to be safe and effective, even in special populations, such as patients with chronic kidney disease and immunocompromised, pediatric, pregnant, and older individuals [13,14,15]; however adverse events of variable significance, mainly due to autoimmune phenomena, have been reported [16,17,18,19,20,21,22,23,24]. Among these side effects, pain at the injection site, fatigue, headache, nausea and low-grade fever were the most common, while others, such as anaphylaxis, thrombosis with thrombocytopenia syndrome, Guillain-Barré syndrome, myocarditis and pericarditis, autoimmune-like hepatitis, and thrombosis of various veins, though severe, proved to be extremely rare [16,17,18,19,20,21,22,23,24].

Liver cirrhosis is followed by a profound immune dysfunction characterized by alterations in innate (decreased complement activity, reduced chemotaxis, and phagocytosis) and adaptive immunity (decreased memory cells, CD4 helper cells, T cell exhaustion) which leads to an inadequate immune response against a wide range of pathogens [25,26]. The pathogenesis of what is known as ‘cirrhosis-associated immune dysfunction’ resides mainly in impairment of the hepatic reticulo-endothelial system, defective protein production, blood cell dysfunction, and systemic inflammation that is related to hepatocyte destruction. [25]. The aforementioned defect in adaptive immunity most probably explains the hypo-responsiveness to various vaccines, such as the pneumococcal, influenza, hepatitis A and B, tetanus, diphtheria, and pertussis and herpes zoster ones among PWLC [27,28,29,30,31,32].

This review will focus on providing available data on safety and immunogenicity of available anti-SARS-CoV-2 vaccines in patients with liver cirrhosis.

## 2. mRNA Vaccines

Although the idea of using messenger RNA (mRNA) for the production of specific proteins has been present since the 1980s, it was only because of the COVID-19 pandemic that this technology was developed and widely used, in a relatively short time [33]. The mRNA-based vaccines introduce an mRNA molecule that is capable of coding a part of the SARS-CoV-2 spike protein [34]. To date, two mRNA vaccines are approved against SARS-CoV-2, namely the BNT162b2 from Pfizer, BioNTech and the mRNA-1273 from Moderna [35,36]. These novel vaccines have proved to be safe and effective in a large variety of patients, including individuals with obesity, chronic lung disease, diabetes mellitus and cardiac disease, among others [35,36].

Only a small number of patients with chronic liver diseases (CLD) and/or cirrhosis, were included in phase I-III clinical trials of mRNA vaccines; therefore a lot of concern was raised regarding their safety and efficacy in these patients [17]. This concern was further increased after the association of mRNA vaccination with liver-related adverse events, with the most important being autoimmune-like hepatitis; this side effect was not attributed exclusively to mRNA vaccination, but in the largest so far case series by Efe C et al., 67 out of 87 patients were mRNA-vaccinated [19,37,38,39]. The efficacy of mRNA vaccines in PWLC was proven quite early in the pandemic, through a study from John B et al., showing that even one dose of mRNA vaccination was associated with a 64.8% reduction in COVID-19 infections and 100% protection against hospitalization or death due to COVID-19; these percentages rose up to 78.6% and 100% respectively after 2nd dose [9]. Interestingly enough, the effect of anti-SARS-CoV-2 vaccination was lower in patients with decompensated cirrhosis, even though the number of patients and events among decompensated PWLC in this study were low [40]. Likewise, in a more recent study by Ge J et al., anti-SARS-CoV-2 vaccination was associated with a 66% all cause mortality in PWLC suffering from COVID-19 [9], while in another study by John BV et al., comparing vaccine-induced versus infection-induced immunity, in a large cohort of PWLC, vaccination led to a reduced risk of developing symptomatic SARS-CoV-2 infection; notably, when symptomatic, the infection had a lower risk of being moderate or severe [41]. Most importantly, in another study by John BV et al., a third dose of a mRNA vaccine was shown to significantly ameliorate immunity against SARS-CoV-2, leading to an even greater reduction in overall and symptomatic SARS-CoV-2 infection, as well as COVID-19-related mortality [42]. This effect was more evident among patients with compensated liver cirrhosis and those receiving the BNT162b2 vaccine.

As far as antibody responses are concerned a variety of immunological responses have been investigated. Regarding post-vaccination antibody responses, Thuluvath P., et al., showed that the vast majority of PWLC had adequate antibody responses, with 91% of the patients in this study having received mRNA-based vaccines; interestingly, vaccination with the viral vector-based Ad.26.COV2.S vaccine was associated with poor immune response in multivariable analysis [43]. In the first ever study measuring neutralizing antibodies in a small cohort of PWLC, vaccinated exclusively with mRNA vaccines, Bakasis AD et al., showed that the presence of liver cirrhosis was not associated with statistically significant differences in total or neutralizing antibodies when compared with patients with CLD and healthy individuals; three months post-vaccination all patients exhibited lower levels of total and neutralizing antibodies, again with no differences between patients with and without liver cirrhosis [44].

In contrast with the preliminary studies, the subsequent studies of mRNA vaccinated PWLC showed contradictory results. In agreement with the aforementioned studies, Ruether D et al., showed that all 53 PWLC developed adequate anti-SARS-CoV-2 antibodies, with no differences in antibody-titers when compared with healthy controls [45]. On the contrary, Willuweit K et al., showed that even though 96% of their PWLC developed anti-SARS-CoV-2 antibodies after two doses of the BNT162b2 vaccine, their levels were lower than those of healthy controls, showing furthermore a rapid and significant decline [46]. Likewise, Al-Dury S et al., Giambra V et al., and Iavarone M et al., showed suboptimal antibody responses in PWLC when compared with controls, even though none of these studies showed higher rates of SARS-CoV-2 infection post-vaccination or COVID-19 severity in PWLC [47,48,49]. Interestingly, two of these studies showed defective T-cell reactivity, while, in the third one, T-cell responses were similar to controls, further complicating the scenery of post mRNA-vaccination T-cell responses in PWLC [47,48,49]. In the study by Giambria V et al., a booster dose of the BNT162b2 vaccine was administered; after that third dose both humoral and cellular responses improved [47]. In another, very interesting study by Chauhan M et al., PWLC and those with liver transplantation with low anti-SARS-CoV-2 after the initial vaccination received a booster dose [50]. A total of 18 PWLC were included in the study, with 12 of them having received mRNA vaccines as initial vaccination and 6 the Ad.26.COV2.S vaccine. Seventeen of these patients received mRNA as a booster vaccine and 1 the Ad.26.COV2.S vaccine, with 12 of them having good antibody responses [50]. The most important studies concerning mRNA-based vaccination against SARS-CoV-2 in PWLC can be found in Table 1.

Overall, mRNA vaccines seem to lead to adequate antibody responses, though in lower titers than healthy individuals; moreover PWLC exhibit faster antibody waning, reflecting their cirrhosis-related immune dysfunction. The different results found in the abovementioned studies most probably reflect differences in methods of antibody measurement and cut-off points rather than the real outcome of mRNA vaccination in these patients. In any case, booster doses seem to overcome any immune deficiency leading to higher antibody production.

Despite their differences regarding T- and B- cell responses adverse events of mRNA-based vaccines were rare and non severe in all of the above-mentioned studies. Among them, local pain, fatigue and low-grade fever were the most common, with severe adverse events being extremely rare. Moreover, no differences among patients with cirrhosis and healthy controls were noted, proving the safety of mRNA vaccination among PWLC [43,44,45,46,47,48,49].

## 3. Viral Vector-Based Vaccines

Viral vector vaccines use an unrelated harmless virus, more often an adenovirus, to deliver genetic material which can be transcribed by the recipient’s host cell as mRNA coding for a desired protein to elicit an immune response [51,52]. These vaccines are further divided in two types: non-replicating and replicating; non-replicating use replication-deficient viral vectors to deliver genetic material of a particular antigen to the host cell to induce immunity against the desired antigen, while replicating ones produce new viral particles in the cells they enter.

Currently available anti-SARS-CoV-2 adenovirus vector-based vaccines include Ad.26.COV2.S by Johnson and Johnson (Janssen) along with Beth Israel Deaconess Medical Center, ChAdOX1-nCOV by Oxford-AstraZeneca, Gam-COVID-Vac and Sputnik Light by Gamaleya Research Institute of Epidemiology and Microbiology, and Ad5-nCoV-S vaccine by CanSino Biologics, making viral vector-based vaccines the most commonly used anti-SARS-CoV-2 vaccines worldwide [53,54]

Adenovirus vector-based vaccines have proven to be safe and effective in the general population [17,18,51]. However, only a handful of studies have evaluated their efficacy and adverse effects in PWLC (Table 2). In the largest of them, a cross-sectional observational study by Singh et al., comprising 231 PWLC, vaccinated with Oxford AstraZeneca ChAdOX1-nCOV vaccine (with 134 of them receiving 2 doses), COVID-19 infection was documented in 3.9% of those who received at least one and in 3.7% of those who received two vaccine doses, with no patients requiring oxygen supplementation or hospitalization [55]. Vaccination related systemic adverse events were rare, with the most common being low grade fever and myalgia in 15.2% and 6.5% of the patients respectively [55]. Lastly, seroconversion was observed in 81 out of 88 patients sampled, with no differences between compensated and decompensated PWLC. Likewise, in a study from Ivashkin V et al., comprising 89 patients with liver cirrhosis vaccinated with Gam-COVID-Vac, vaccination led to a lower rate of SARS-CoV-2 infections and COVID-19 or liver-related mortality, with no significant adverse events [56], while in a study by John BV et al., vaccination with a single dose Ad.26.COV2.S vaccine was modestly effective against overall COVID-19, but showed good effectiveness against severe/critical COVID-19; with no statistical significance when compared with mRNA-based vaccines [57].

In contrast with the study from John BV et al., antibody responses after adenovirus-based vaccines were poorer when compared with mRNA-based vaccines, especially in patients with decompensated cirrhosis, in two other studies [43,58]. More specifically, Thuluvath P., et al. showed poorer antibody responses in patients with cirrhosis receiving the Ad.26.COV2.S vaccine when compared with those receiving the mRNA vaccines; however only seven PWLC received the Ad.26.COV2.S vaccine in this study [43]. Likewise, in a study by Kulkarni A et al., patients with decompensated cirrhosis showed suboptimal humoral and cellular immune responses after ChAdOx1 vaccination [58]. More specifically, 34% of patients with decompensated cirrhosis were non-responders, while CD4-naïve, CD4 effector, B- and B-memory cells were lower in patients with decompensated cirrhosis [58]. In agreement with the previous studies, no severe adverse events were recorded, confirming the safety of viral vector-based vaccines in PWLC.

Overall, the results from these studies show antibody responses similar to that expected from previous experience with vaccination in PWLC, such as hepatitis B or pneumococcal vaccines, with adequate levels of seropositivity but lower levels of serum antibodies [27]; possible differences in various studies most probably reflect endpoints used in each one. Fortunately, viral vector-based vaccines were successful in protecting from severe infection in all studies irrespective of serum antibody levels.

## 4. Whole Virion Vaccines

Inactivated whole virion vaccines use an inactivated (by chemicals, heat, or radiation) form of the disease-causing virus, incapable of causing disease but capable of generating an immune response [51,52]. To date, three different inactivated COVID-19 vaccines are commercially available; namely, the Sinopharm BBIBP-CorV and WIBP-CorV vaccines, the Sinovac PiCoVacc vaccine, and the Bharat Biotech BBV152 COVAXIN vaccine [59].

Various studies have shown that inactivated anti-SARS-CoV-2 vaccines are both safe and effective in healthy adults [53,60,61]. On the other hand, data concerning PWLC are rather scarce, with only a few studies evaluating the impact of these vaccines in COVID-19-related hospitalization and mortality. In one of the largest studies, by Diaz LA et al., comprising 2050 PWLC and COVID-19, vaccination led to substantial decreases in hospitalization rates; in this study 79.4% of the whole cohort was vaccinated with the PiCoVacc vaccine [62].

In another large study, a prospective, multicenter study, by Wang J et al., a total of 553 patients with cirrhosis received two doses of inactivated whole-virion COVID-19 vaccines and adverse events and neutralizing antibodies were assessed [63]. Overall vaccination was found to be safe and well tolerated for patients with both compensated and decompensated liver cirrhosis. The most common local adverse reaction was injection site pain followed by swelling and erythema and the most common systematic adverse reaction was fatigue and fever; all adverse events were mild and resolved spontaneously [63]. Interestingly, decompensated cirrhosis was correlated with vaccine hyporesponsiveness, with Child-Pugh grades B and C being independent risk factors for absence of neutralizing antibodies [63]. In another prospective, multicentered, open label, study by Ai J et al., 153 patients with cirrhosis received two doses of inactivated whole virion SARS-CoV-2 vaccines; the overall positive rate of neutralizing antibodies was significantly lower when compared with healthy participants, with similarly low immunogenicity in compensated and decompensated patients (78.9% and 76.7% respectively). Adverse events among vaccine recipients were mild and transient, with injection site pain being the most common; only three patients exhibited severe transaminasemia post-vaccination, with one of them being judged as severe and needing hospitalization [64]. Likewise, in another prospective observational study, by Chen Z et al., comparing healthy controls with patients with severe liver disease, PWLC showed inferior antibody responses that also waned faster; no statistically significant differences were found among patients with compensated and decompensated liver cirrhosis [65]. Another interesting finding of this study was the fact that the overall incidence of adverse events within seven days post-vaccination in PWLC was significantly higher than that of healthy controls (33,3% vs. 12,0%); the majority of these adverse events were mild and non-severe [65]. The most important studies regarding the outcomes of whole virion vaccines in PWLC can be found in Table 3.

Overall, vaccination with whole virion vaccines seem to show suboptimal antibody production in PWLC, especially in those with decompensated disease, when compared with other available vaccines; however data on the clinical impact of this (with regard to COVID-19 related mortality and need for hospitalization) are largely lacking.

## 5. Protein Subunit Vaccines

Protein subunit vaccines are based on the use of an antigenic protein part, commonly combined with an adjuvant, to enhance immunogenicity [66,67]. Multiple protein subunit vaccines against SARS-CoV-2 are currently under trial [68], even though up to date only two protein subunits vaccines are approved for use, namely the NVX-CoV2373 vaccine, sold under the names of Nuvaxovid from Novavax, and Covovax from Serum Institute of India [69]. These vaccines seem to elicit both B-lymphocyte and T-lymphocyte immune responses to the S protein of SARS-CoV-2, while they seem to offer protection against most variants of the virus [70,71]. Moreover, the protein subunit vaccines have showed minimal side effects in clinical trials; the observed side effects were mainly local, such as injection site pain and swelling, redness, and pruritus [70,71,72].

Unfortunately, up until now, no trials exist regarding the use of these vaccines in PWLC, so no safe consumptions can be reached, even though the safety and efficacy profile of these vaccines make them good candidates for use in this population.

## 6. Discussion

The use of anti-SARS-CoV-2 vaccines has proven to be the most efficient course of action against the COVID-19 pandemic. Various vaccine types, namely mRNA, inactivated or weakened virus, viral vector, and protein subunit ones, have been approved and are widely used. These vaccines have shown high efficacy and safety in both high risk patients and healthy individuals. Even though only a few PWLC were included in phase II-III studies of these vaccines, real world data are constantly increasing, showing that all vaccine types are extremely safe, with only minor adverse effects, such as pain at the injection site, fatigue, or low-grade fever.

Antibody responses in these patients are still debatable, with various results in various studies. Overall patients with compensated cirrhosis seem to follow the antibody responses of healthy individuals, especially after mRNA vaccination, with inferior results after whole virion and viral-vector based vaccination. On the other hand, decompensated PWLC seem to have significantly weaker T- and B-cell responses, with lower total and neutralizing antibodies. Moreover, these patients seem to diminish their antibody serum titers faster than healthy individuals or non-cirrhotic patients with CLD. These results come in agreement with previous studies regarding antibody responses in other vaccines, such as those for hepatitis A, B, or pneumococcal disease [27,28,29,30,31,32]. Interestingly, mRNA vaccination gives slightly better responses than other vaccines; most probably due to the way these vaccines lead to antibody production [73]. According to some studies immunogenicity of adenovirus based vaccines might be subordinate due to pre-existent immunity against the viral vector (especially when that is Ad5) [[52],[74], while protein subunit and inactivated viral vaccines trigger inadequate CD8+ T lymphocyte responses and need both adjuvant molecules and repetitive doses, to achieve immunogenicity [74]. Perhaps the answer lies in booster doses, since a handful of studies have shown amelioration of immune responses after a 3rd dose; data however are still scarce regarding PWLC.

Even though T- and B- cell responses seem to be inadequate in PWLC, the clinical output of these findings is still in doubt, since both compensated and decompensated PWLC exhibit low rates of SARS-CoV-2 infection, severe COVID-19, and COVID-19 related mortality, after vaccination, with only a few patients requiring hospitalization or oxygen support (Figure 1).

Of course, in a still ongoing and constantly evolving COVID-19 pandemic, many questions remain. Most of the studies presented in this article concern the Alpha (B.1.1.7) and Delta (B.1.617.2) variants, two variants that have now been substituted with the Omicron (B.1.1.529) variant and its subtypes [75]. This new variant presents a variety of mutations, leading to reduced vaccine effectiveness against mild disease with rapid antibody waning [76,77,78], needing booster doses form either mono- or bivalent vaccines for increased protection [76,79,80,81]. Notably, overall mortality from the omicron strain seems to be lower than the previous ones, however no studies exist regarding this matter in PWLC. In this new landscape of new SARS-CoV-2 variants, booster dose (or even doses) should probably be recommended to PWLC (like in all high-risk patients), even though the number and frequency of these doses is still debatable for both healthy individuals and PWLC. Other questions that should be addressed in large, well designed, long-standing trials are the efficacy of protein subunit vaccines in PWLC, as well as the actual use of antibody responses especially in decompensated PWLC. Last, but not least, the successful use of the new mRNA vaccines and their adequate antibody responses in PWLC might be a good opportunity to reconsider vaccination in this population; mRNA technology could be used to replace vaccines with low immunogenicity in PWLC, such as the pneumococcal or the hepatitis A and B vaccines.

## 7. Conclusions

Overall, anti-SARS-CoV-2 vaccines seem to be safe and effective in PWLC, protecting from severe COVID-19 with only minor adverse events. Questions regarding the clinical significance of antibody responses and booster doses in these patients remain to be answered.

## Figures and Tables

**Figure 1 vaccines-11-00452-f001:**
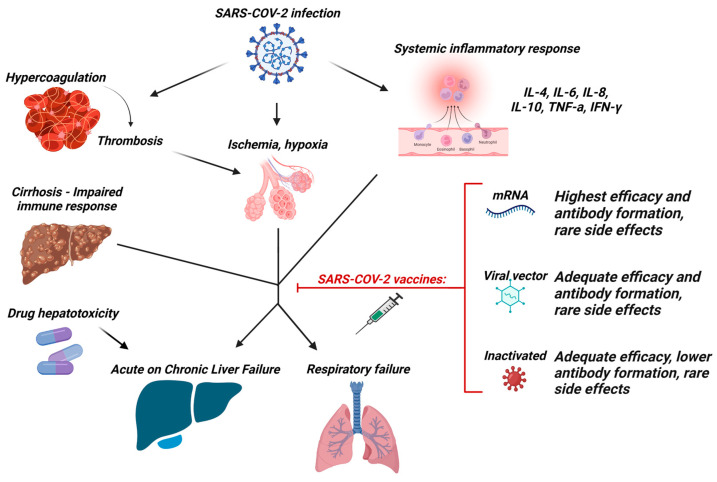
Mechanisms of liver and lung injury caused by SARS-CoV-2 infection in patients with cirrhosis and efficacy of commonly used vaccines. Abbreviations: IL: Interleukin; SARS-CoV-2: Severe acute respiratory syndrome coronavirus 2. Created with BioRender.com. Used under license nr VA24ZJX03J.

**Table 1 vaccines-11-00452-t001:** Most important studies concerning mRNA-based anti-SARS-CoV-2 vaccines in patients with liver cirrhosis.

First Author, Year	Type of Trial	Number of PWLC	Controls	Results
John B.V., 2021 [40]	Retrospective	20,037 with at least one dose of BNT162 mRNA or mRNA1273 vaccine	20,037 unvaccinated PWLC	64.8% reduction in COVID-19 cases and 100% reduction in hospitalizations and SARS-CoV-2-related deaths 28 days post-vaccination. A second dose of an mRNA vaccine further decreased infection rate to 78.6% reduction in COVID-19 cases after 2nd dose. Reduction more evident among those with compensated cirrhosis.
Tuluvath P.J., 2021 [43]	Prospective	79 (10 decompensated)	62 LT recipients and 92 with CLD but no cirrhosis	3 PWLD with no and 15 with suboptimal antibody responses. Cirrhosis not associated with poor antibody responses in multivariable analysis
Bakasis A.D., 2022 [44]	Prospective	38 (13 decompensated)	49 with CLD and 40 healthy controls	Seroconversion rate 97.4% and neutralizing activity 92.1% in PWLC. Cirrhosis not associated with poor antibody responses in multivariable analysis
Iavarone M., 2022 [49]	Prospective	182 (28 with previous SARS-CoV-2 infection, 154 without)	38 healthy (12 with previous SARS-CoV-2 infection, 26 without)	Anti-spike protein S antibody titers statistically significantly lower in PWLC. T-cell responses lower in PWLC, but not statistically significant
John B.V., 2022 [41]	Retrospective	27,131 with 2 doses of mRNA vaccine	634 PWLC who developed immunity after infection	Vaccine-induced immunity better regarding infection susceptibility, symptomatic and moderate/severe critical disease.
John B.V., 2022 [42]	Retrospective	13,041 with 3 doses of mRNA vaccine	13,041 PWLC with 2 doses of mRNA vaccine	80% reduction in infections, symptoms, and severe disease, and 100% reduction in death rate with a 3rd dose. Better results with BNT162b, and in compensated cirrhosis.
Giambra V., 2022 [47]	Prospective	151 vaccinated with BNT162b2	117 healthy controls vaccinated with BNT162b2	Delay in B-cell and lack of prompt T-cell response in PWLC. No difference in breakthrough infections among PWLC and controls.
Ruether D.F., 2022 [45]	Prospective	53 (91.6% mRNA vaccinated)	138 LT recipients (87.7% mRNA vaccinated) and 52 controls (75% mRNA vaccinated)	Seroconversion achieved in 100% of PWLC, lower spike-specific T-cell responses than controls, higher than LT recipients.
Willuweit K., 2022 [46]	Retrospective	110, vaccinated with 2 doses of BNT162b2	80 healthy controls vaccinated with 2 doses of BNT162b2	No significant difference in seroconversion rates, but lower antibody titers in PWLC. Rapid and significant decrease in antibody titers in PWLC
Al Dury S., 2022 [48]	Prospective	48, all mRNA vaccinated	39 healthy controls	68% of PWLC with undetectable anti-SARS-CoV-2 T-cell reactivity after 1st dose; 36% after 2nd, significantly lower than controls. Likewise for anti-SARS-CoV-2 antibody titers. Lower antibody levels and worse T-cell reactivity in advanced cirrhosis.

Abbreviations: COVID-19: Coronavirus disease 2019; LT: Liver transplantation; mRNA: Messenger RNA; PWLC: Patients with liver cirrhosis; SARS-CoV-2: Severe acute respiratory syndrome coronavirus 2.

**Table 2 vaccines-11-00452-t002:** Most important studies concerning viral-vector anti-SARS-CoV-2 vaccines in patients with cirrhosis.

Name of First Author, Year	Type of Study	Number of PWLC	Number of Controls	Outcome
Singh A., 2022 [55]	Cross sectional, observational	97 with one ChadOx1-nCOV vaccine dose, 134 with two doses	N/A	92,1% with detectable antibodies (6,8% low, 77,3% moderate and 8% high titers). No differences in antibody responses among compensated and decompensated PWLC
Ivashkin V., 2022 [56]	Retrospective cohort	89 vaccinated with Gam-Covid-Vac	148 unvaccinated PWLC	Vaccine efficacy of 69.5% against symptomatic, 100% against severe COVID-19, and 100% against COVID-19 associated death. Higher overall mortality in unvaccinated group. No significant differences in liver-related mortality, incidence of liver decompensation and bleeding esophageal varices between two groups
John B.V., 2022 [57]	Cohort	94 vaccinated with Ad.26.COV2.S	1089 mRNA vaccinated and 727 unvacinated PWLC	Ad.26.COV2.S efficacy of 64% against COVID-19, with 72% against severe/critical COVID-19. no statistically significant differences between viral vector and mRNA vaccines.
Thuluvath P.J., 2021 [43]	Prospective	7 vaccinated with Ad.26.COV2.S	41 mRNA1273 vaccinated and 31 BNT162b2 vaccinated PWLC, 62 LT vaccinated patients and 92 vaccinated patients with liver diseases, non-cirrhotic	JNJ-78435735 vaccination associated with poor immune response in multivariable analysis
Kulkarni A.V., 2022 [58]	Prospective	90 ChAdOx1, 23 BBV152	60 healthy controls, 50 patients with liver diseases, non-cirrhotic, 17 LT patients	No differences in antibody responses among healthy controls, non-cirrhotics and compensated PWLC. 34% of decompensated PWLC non-responders. 34% with decompensated cirrhosis non-responders.CD4-naïve, CD4-effector, and B-memory cells lower in decompensated PWLC

Abbreviations: COVID-19: Coronavirus disease 2019; LT: Liver transplantation; mRNA: Messenger RNA; N/A: Not available; PWLC: Patients with liver cirrhosis.

**Table 3 vaccines-11-00452-t003:** Most important studies concerning whole virion anti-SARS-CoV-2 vaccines in patients with cirrhosis.

Name of First Author, Year	Type of Study	Number of PWLC	Number of Controls	Outcome
Wang J., 2022 [62]	Prospective, multicenter	340 PiCoVacc, 151 BBIBP-CorV and 62 WIBP-CorV	N/A	71,6% with compensated and 66,6% with decompensated cirrhosis had positive rates of COVID-19 neutralizing antibodies
Ai J., 2021 [63]	Prospective, multicenter	153 vaccinated with PiCoVacc, BBIBP-CorV or WIBP-CorV	284 with CLD, non-cirrhotic,144 healthy controls	Statistically significantly lower neutralizing antibodies in all CLD patients when compared with healthy controls. No differences between non-cirrhotic, compensated and decompensated cirrhotic patients with CLD
Chen Z., 2022 [64]	Prospective, observational	127 vaccinated with BBIBP-CoV or PiCoVacc	142 healthy controls and 65 patients with HCC	Seropositivity rate of anti-SARS-CoV-2 antibodies high in PWLC and patients with HCC. Lower detection levels of neutralizing antibodies in PWLC and patients with HCC. Lower antibody responses in CTP B and C compared with A.

Abbreviations: COVID-19: Coronavirus disease 2019; CTP: Child-Turcotte-Pugh; CLD: Chronic liver diseases; HCC: Hepatocellular carcinoma; N/A: Not available; PWLC: Patients with liver cirrhosis.

## Data Availability

Not applicable.

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
