# Peer review of "Efficacy, Safety and Immunogenicity of Anti-SARS-CoV-2 Vaccines in Patients with Cirrhosis: A Narrative Review"

_vaccines, 2023, doi:10.3390/vaccines11020452_

Round 1

Reviewer 1 Report

This is an interesting review. However, it is a summary of references. How to explain the contradictory efficacy and what is the future direction? There are mistyping in the manuscript. For example, page 2, line 97,enoughwhen should be enough when.

Reviewer 2 Report

In manuscript Vaccines-2178248, the authors present an overview of our recent understanding of the anti-SARS-CoV-2 vaccines in patients with cirrhosis. This is a manuscript submitted for publication as Narrative review.

Patients with chronic liver diseases, including liver cirrhosis, are considered to be at high-risk for severe COVID-19. Importantly, owing to defect in adaptive immunity, patients with liver cirrhosis are expected to be hypo-responsiveness to various vaccines. These patients thus constitute a special population for vaccine validation, yet the inclusion of this population in vaccine trials is still limited.

In this review, the authors discussed the available data in regards to the safety, efficacy and duration of protection of the vaccination against anti-SARS-CoV-2 in patients with liver cirrhosis.

This topic is of interest in the field and quite novel. This important consideration is clearly discussed by the authors: divided by sections corresponding to each type of vaccine, with sum up of data from the clinical trials presented in Tables. Nevertheless, there is still a number of unsolved questions. In this regard, some specific points should be addressed by the authors for improvement of this Review Article, as follows.

Specific points.

1-      The inclusion of a general schematic representation of the features of SARS-CoV-2 diseases and vaccine trials in patients with liver cirrhosis would help to better highlight the salient conclusions and still-unsolved questions.

2-      The authors should include a section at the end on the vibrant still-unsolved questions on this topic, that will be the challenges in the coming years.

3-      The authors should further present the current knowledge (albeit limited) in regards to the variant of concerns of SARS-CoV-2.

4-      The authors should open discussion on the related knowledge for other viruses and/or how the recent findings on anti-SARS-COV-2 vaccine can be a framework and/or pave the way to better our understanding on other viruses.

Minor points:

1- Some unnecessary spaces, and typos should be corrected.

2- Some vaccines should be better defined.

3- The authors should check and include the very recent publications/update on the vaccine trial.

Reviewer 3 Report

This review describes anti-SARS-CoV-2 vaccines administering for patient with liver cirrhosis. It covers all types of vaccines including mRNA, viral vector-based, whole virion, and protein subunit vaccines, and reviews efficacy and adverse effects of vaccine administration to patients with liver cirrhosis. The review is timely, up-to-date and informative for many readers.

Minor comment

A space might be needed between “dose” and “both” (P.3, Line 129), and between “response” and “adverse” (P.5, Line 143).

Round 2

Reviewer 1 Report

The authors have corrected the disadvantages. 

  •  

Reviewer 2 Report

The authors have addressed all my previous comments